# Adoption of climate-resilient groundnut varieties increases agricultural production, consumption, and smallholder commercialization in West Africa

Martin Paul Jr Tabe-Ojong [1] ✉, Jourdain C. Lokossou [2], Bisrat Gebrekidan[3] & Hippolyte D. Affognon[4]

As part of the climate-smart agriculture approach, the adoption of climate-resilient crop varieties has the potential to build farmers' climate resilience but could also induce agricultural transformation in developing nations. We investigate the relationship between adoption of climate-resilient groundnut varieties and production, consumption, and smallholder commercialization using panel data from Ghana, Mali, and Nigeria. We find adoption of climate-resilient groundnut varieties to increase smallholder production, consumption, and commercialization. The biggest adoption impact gains are observed under the sustained use of these climate-resilient varieties. We show that adoption benefits all households, but the biggest gains are found among smaller producers, suggesting that adoption is inclusive. Furthermore, we provide suggestive evidence that yield increases could explain commercialization, although household consumption also matters. We conclude that adoption of climate-resilient groundnut varieties can at least partially reduce production constraints and promote smallholder consumption and commercialization, with implications for agricultural transformation.

Smallholder commercialization has been at the forefront of many policy debates as a pathway to reducing poverty in many developing countries[1]. Considering these debates, many governments have established commercialization initiatives to drive agricultural transformation. However, their success depends to a large extent on agricultural productivity[2]. Agricultural productivity growth is a crucial ingredient for economic diversification and development[3,4]. Still, agricultural productivity growth continues to be low in Sub-Saharan Africa, lagging behind other regions of the world[2,5]. The Green Revolution, the impetus behind numerous strides in increasing agricultural production and productivity, continues to be replicated in many countries in the region[6,7]. Key in these efforts has been the development and dissemination of high-yielding and disease-resistant crop varieties[8]. Given the extended dry seasons common in arid and semi-arid zones, some of these crop varieties are climate-resilient, with the ability to withstand extreme weather events and build climate resilience[9–11]. Climate-resilient crop varieties are a critical part of the climate-smart agriculture (CSA) approach with the potential to offer the triple wins of increasing productivity with ensuing welfare implications, building resilience to climatic shocks and reducing the emission of greenhouse gases[12].

We examine the relationship between adoption of climate-resilient groundnut varieties and production, consumption, and

[1]Development Strategy and Governance Unit, International Food Policy Research Institute (IFPRI), Cairo, Egypt. [2]Department of Agri-Food Economics and Consumer Sciences, Faculty of Food Science and Agriculture, Laval University, Quebec, Canada. [3]Institute for Food and Resource Economics, University of Bonn, Nußallee 19-21, 53115 Bonn, Germany. [4]West and Central African Council for Agricultural Research and Development (CORAF), Dakar, Senegal. ✉e-mail: tabeojongmartinpaul@gmail.com

smallholder commercialization, taking advantage of extraordinarily rich farm-level data in three West African countries (Ghana, Mali, and Nigeria) from 2017–2019. Many of the farms surveyed are cultivated by small-scale farmers who produce groundnut to satisfy their household food demand but also sell some output in markets, potentially enabling them to escape the poverty trap of semi-subsistent agriculture. Groundnut is an important food and cash crop in Sub-Saharan Africa[13]. This important legume has been associated with poverty reduction through increasing household income[14] and offering benefits beyond food and cash, as it can help in the synthesis of atmospheric nitrogen, which in turn helps in improving soil fertility. This may reduce the use of inorganic fertilizers, as the legume crop itself improves soil fertility[15]. As a food crop, groundnut has multiple nutritional properties, containing both protein and fats/oils.

In this study, we use a household fixed effect (FE) estimator and a correlated random effect (CRE) estimator to control for the unobserved heterogeneity associated with the relationship between adoption and production, consumption, and commercialization. We find a positive association between adoption of climate-resilient groundnut varieties and first-order outcomes such as production (as measured by production, production value, and yields) and higher-order outcomes such as consumption and commercialization (as measured by market participation, quantity sold, and sales value). We find that sustained adoption over time (that is, over the three consecutive survey years) increases smallholders' groundnut production, consumption, and commercialization more than one or two years of adoption. Cross-country evidence from Ghana, Mali, and Nigeria demonstrates substantial heterogeneity. Nonetheless, the results are robust to different estimation strategies, variable measurements, and transformations as well as different assumptions about the panel estimator and instrumental variable (IV).

Additional analyses show a positive association between production and commercialization and a negative association between consumption and commercialization. These insights are consistent with the nonseparability of households' production and consumption decisions in the face of imperfect market conditions[16,17]. Even with well-functioning markets, households may keep some production for home consumption[17]. Of course, households in many farming systems will only participate in markets after their household consumption demands are met. Beyond associations at the mean, we perform regressions to determine the association between adoption of climate-resilient groundnut varieties and quantiles of the conditional distribution of commercialization. While adoption benefits all

households, the biggest commercialization gains are observed among small-scale farmers. This important finding suggests that the use of climate-resilient groundnut varieties is inclusive (that is, it does not exclude any category of farmers). Finally, we show that increased production is important for smallholder commercialization, although household consumption also matters.

## Results and discussion
### Summary statistics
We begin with a descriptive summary of some of the key variables of interest (see Table S1 in the supplementary information for summary statistics by year and adoption status). Overall, adopters of climate-resilient groundnut varieties are relatively younger (48 years) and better educated (-4 years of schooling) than nonadopters. Their household size is relatively smaller (10 people) and adopting households are closer to urban markets (11 km on average). About 56% of adopters belong to a producer group and have received more visits from public extension services (on average about 3 visits). Figure 1 displays the kernel density distribution of smallholder production and commercialization, both measured in kilogram (kg) where the vertical lines indicate the mean production and sales of groundnuts and differentiate them by adoption status. As can be seen from the figure, adopters of climate-resilient groundnut varieties produce and sell more quantities of groundnuts than nonadopters. Figure 2 shows adoption of improved groundnut varieties over the three survey years (2017–2019). Adoption is increasing in Mali and Nigeria, but not in Ghana. While this may suggest that seed systems are functioning well in these two countries, it could also be due to seed distributions by nongovernmental organizations, farmer-to-farmer exchanges, government subsidies as well as other (latent) factors.

### Adoption effects on production, consumption, and commercialization
This section presents four sets of empirical results. First, we establish the relationship between adoption of climate-resilient groundnut varieties and production, consumption, and commercialization. Second, we perform cross-country analyses of these relationships in Ghana, Mali, and Nigeria. Third, we link production and consumption with commercialization. Finally, we show the heterogeneous relationship between adoption and commercialization using quantile regressions.

Figure 3 shows the relationship between adoption and production and consumption. Adoption is defined in two ways: as a binary variable

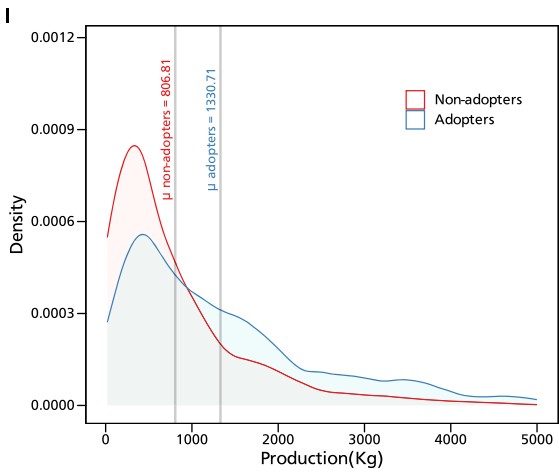
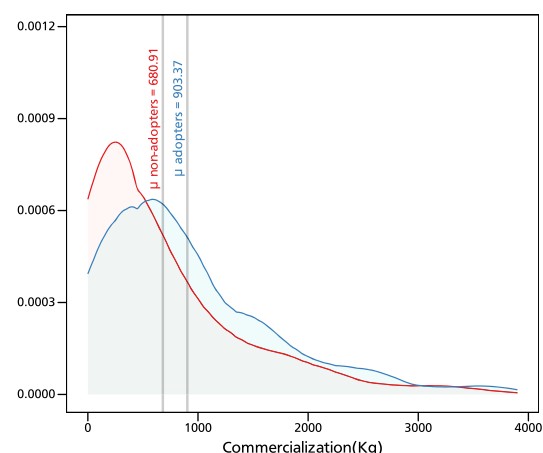

**Fig. 1 | Kernel density distribution of groundnut production and commercialization.** This figure shows the distribution of production and commercialization for adopters and nonadopters. $N = 8604$ observations. While Panel **I** shows the mean difference for groundnut production, panel **II** shows the mean difference for commercialization. Source data are provided as a Source data file.

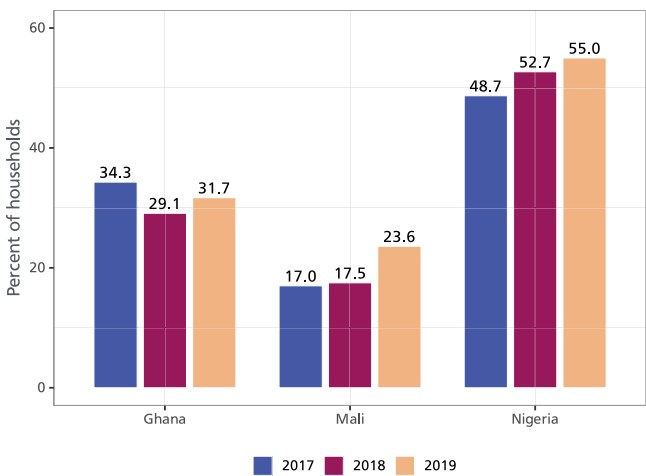

**Fig. 2 | Adoption of climate-resilient groundnut varieties by country, 2017–2019.** Figure 2 shows the proportion of adopters by year and country. $N = 1494$, 2520, and 4590 observations in Ghana, Mali, and Nigeria, respectively. Source data are provided as a Source data file.

(the extensive measure) that captures the transition from nonadoption to adoption; and as a continuous variable that measures the extent (area) of adoption (the intensive measure). For ease of presentation, only these two coefficients are shown in the following figure, but Table S8 in the supplementary information reports the full estimation results. Using the first definition of adoption, a positive association exists between adoption of climate-resilient groundnut varieties and household production and consumption: Adopting climate-resilient groundnut varieties increases yield by about 345 kg/ha and production value by USD 476 (Fig. 3). The extent of adoption also positively affects both yield and production value, although to a lesser extent. This may be due to diminishing returns to area under adoption, possibly signifying a nonlinear relationship.

We also observe a positive association between adoption of climate-resilient groundnut varieties and their consumption by smallholders: adoption increases home consumption by about 213 kg (Fig. 3). Our findings on the positive relationship between adoption and yields are in line with insights from refs. 8,11,13,18, who showed that improved (drought-resistant) seeds are high-yielding, with implications for smallholder commercialization. Overall, these results support the importance of climate-resilient groundnut varieties in increasing crop yields under stress conditions[19], as they help farmers cope with climate shocks and build resilience to climate change.

We also establish a positive relationship between adoption (both extensive and intensive measures) and commercialization (market participation, quantity sold, and sales value). Given the presence of zeros in commercialization outcomes indicating no sales, we transform quantity sold and the sales value using the inverse hyperbolic sine transformation, which efficiently manages zeros[20]. This transformation is akin to a log transformation but allows for observations with zeroes and negative values. We use the household FE and CRE estimators. The estimated coefficients are very similar for both, pointing to the robustness of the findings. We find a positive and significant association between both measures of climate-resilient groundnut variety adoption and commercialization (Fig. 4). Adoption versus nonadoption leads to increases of 5–6% in market participation, 54–59% in quantity sold, and 53–57% in sales value, while the extent of adoption leads to increases ranging from 3–4% in market participation, 37–41% in quantity sold, and 35–39% in sales values (Fig. 4). Similar findings have been reported in Malawi, where improved groundnut varieties with ancestry from the genebank of the International Crops Research Institute for the Semi-Arid Tropics (ICRISAT) have been shown to increase market participation[21]. We also present additional insights

from the pooled FE-OLS model about the relationship between adoption, production, and commercialization in Figures S1 and S2 in the supplementary information.

We next show that sustained adoption—defined as continuous and consecutive adoption of the climate-resilient groundnut varieties over the three survey years—is more effective in enhancing smallholders' groundnut production, consumption, and commercialization than one or two years of adoption (Fig. 5). More importantly, the impact magnitudes are multiples of the previous estimates of production, consumption, and commercialization using the extensive measure of adoption. Our findings corroborate those of ref. 13, who showed greater poverty reduction effects for households that adopt climate-resilient groundnut varieties on a sustained basis.

**Cross-country heterogeneity in adoption impacts**

Cross-country analyses help us understand the production, consumption, and commercialization effects of groundnut adoption in Ghana, Mali, and Nigeria. Figure 6 illustrates significant heterogeneity across the three countries, with the strongest yield effects observed in Ghana and Nigeria. For effects on commercialization, only Nigeria exhibits a statistically significant difference. Country-specific factors could explain the observed cross-country heterogeneity. However, these results could also reflect the different household characteristics of each country. For example, Nigeria is the largest producer and exporter of groundnuts in West Africa[22], so Nigerian producers may have access to more diversified markets and conditions under which farmers may receive higher and more favorable prices for their output. This could be a particularly strong incentive for groundnut commercialization, especially after households satisfy home consumption. This hypothesis is consistent with the nonseparable agricultural household model whereby households only approach markets as sellers when their household food demands are met[18].

Since households' production and consumption decisions are closely related and possibly nonseparable[16,17], we run some additional regressions. It is intuitive that an increase in yield arising from adoption of climate-resilient varieties could drive commercialization, but household consumption is also important. Figure 7 shows a positive association between yield and commercialization, giving credence to the claim that the former could explain the commercialization impacts. The negative association between consumption and commercialization further bolsters the insight that households may only participate in markets when their household food demands are met. Thus, while increasing production could drive farmers to markets, home consumption could reduce aspects of commercialization, since households rely on groundnut as a key nutritious food. These results can again be explained by the nonseparable agricultural household model with missing markets[23]. The key insight is that production, consumption, and ultimately market participation decisions are highly interrelated. As yield increases, households will participate in markets only after their household food demands are met. This is especially true for a legume like groundnut, which contributes immensely to the nutritional basket of smallholder households[12]. Ascertaining yield increases seems to be an important pathway to ensuring smallholder commercialization, and climate-resilient groundnut varieties could be a crucial entry point. Farmers' use of climate-resilient crops is also important for building their resilience to climate change and extreme weather events[19].

In terms of impact heterogeneity, adoption benefits households in all quantiles of the conditional distribution of commercialization. However, the biggest gains are observed for farmers who adopt at smaller scales (Fig. 8). Notwithstanding, this finding suggests that adoption is inclusive and may spur development in rural communities through smallholder commercialization.

Taken together, our findings highlight the importance of boosting adoption of climate-resilient crop varieties as a pathway to agricultural

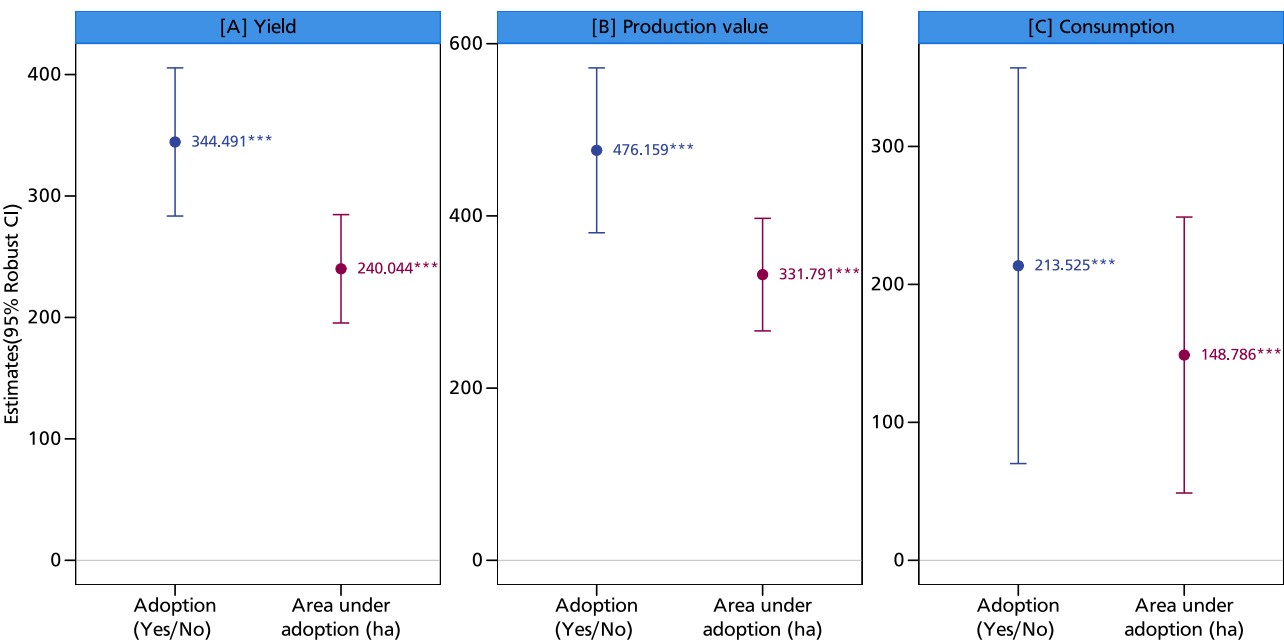

**Fig. 3 | 2SLS estimates of the impact of groundnut adoption on production and consumption.** This figure displays coefficients along with their corresponding 95% confidence intervals as error bars. Panel **A** shows the yield effects of climate-resilient groundnut varieties, panel **B** the production value effects, and panel **C** the consumption effects. The coefficients are estimated using the two-stage least squares regression approach with $N = 8604$ observations. The presence of an asterisk (*) above a coefficient indicates that the coefficient is statistically different from zero at a predetermined level of significance (***$p < 0.01$, **$p < 0.05$, *$p < 0.1$).

Statistical tests are two-sided $t$-tests. Full models are reported in Table S8 in the supplementary information. The models are estimated with additional controls such as age and education level of the household head, dependency ratio, gender of the household head, household size, cooperative membership, training, access to public and private extension, access to both cash and in-kind credit, distance to nearest urban and village market, crop rotation, mixed cropping, labor, market price, input costs, area of cultivation, off-farm income, and soil type. Source data are provided as a Source data file.

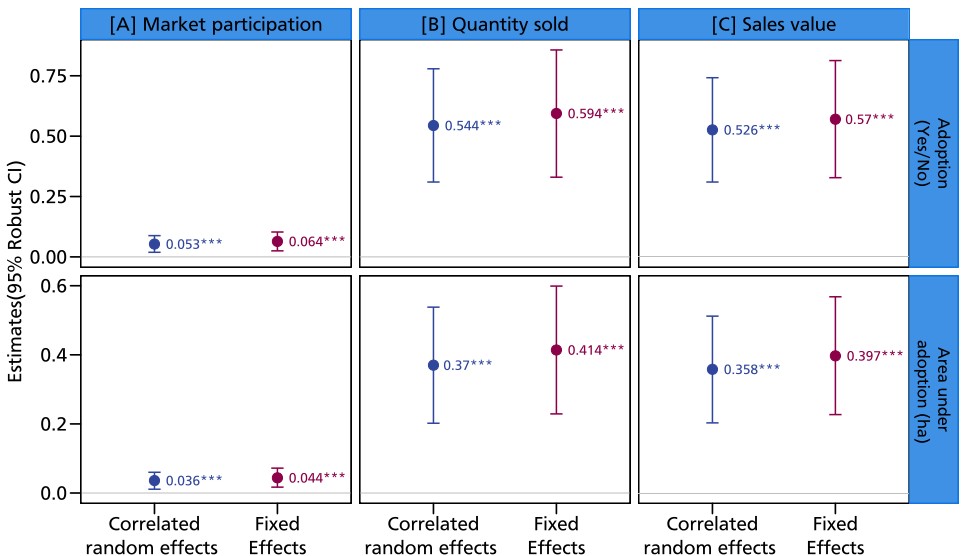

**Fig. 4 | 2SLS estimates of the impact of groundnut adoption on commercialization.** This figure displays coefficients along with their corresponding 95% confidence intervals as error bars. Panel **A** shows the impacts of adoption of climate-resilient groundnut varieties on market participation, panel **B** shows the impacts on the quantity of groundnut sold, and panel **C** the sales value of the groundnut sold. The coefficients are estimated using the two-stage least squares regression approach with $N = 8604$ observations. The presence of an asterisk (*) above a coefficient indicates that the coefficient is statistically different from zero at a predetermined level of significance (***$p < 0.01$, **$p < 0.05$, *$p < 0.1$). Statistical tests

are two-sided $t$-tests. Full models are reported in Tables S6 and S7 in the supplementary information. The models are estimated with additional controls such as age and education level of the household head, dependency ratio, gender of the household head, household size, cooperative membership, training, access to public and private extension, access to both cash and in-kind credit, distance to nearest urban and village market, crop rotation, mixed cropping, labor, market price, input costs, area of cultivation, off-farm income, and soil type. Source data are provided as a Source data file.

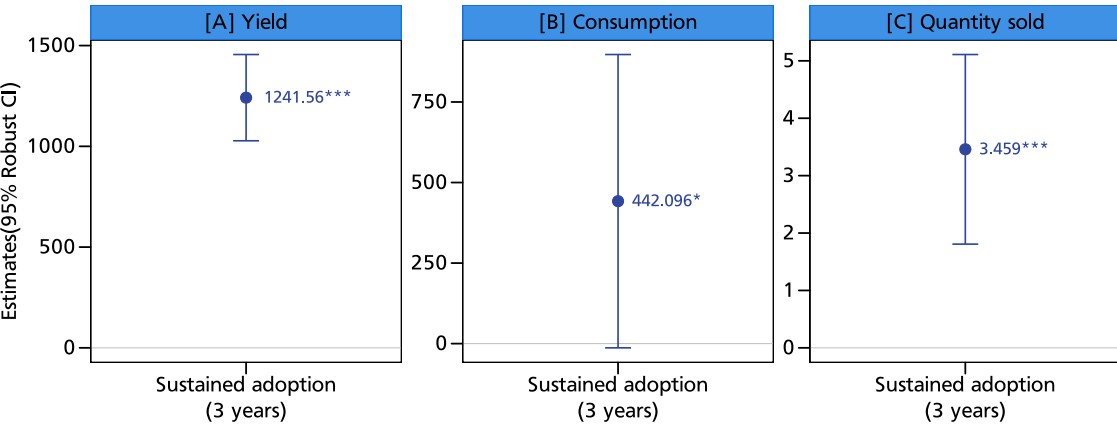

**Fig. 5 | Association between sustained adoption and production, consumption, and commercialization.** This figure displays coefficients along with their corresponding 95% confidence intervals as error bars. Panel **A** shows the impact of sustained adoption of climate-resilient varieties on yields, panel **B** shows the sustained impacts on consumption and panel **C** shows the sustained impacts on the quantity sold. The coefficients are estimated using the two-stage least squares regression approach with $N = 8604$ observations. The presence of an asterisk (*) above a coefficient indicates that the coefficient is statistically different from zero at a predetermined level of significance (***$p < 0.01$, **$p < 0.05$, *$p < 0.1$). Statistical tests are two-sided $t$-tests. Full models are reported in Table S9 in the supplementary information. The models are estimated with additional controls such as age and education level of the household head, dependency ratio, gender of the household head, household size, cooperative membership, training, access to public and private extension, access to both cash and in-kind credit, distance to nearest urban and village market, crop rotation, mixed cropping, labor, market price, input costs, area of cultivation, off-farm income, and soil type. Source data are provided as a Source data file.

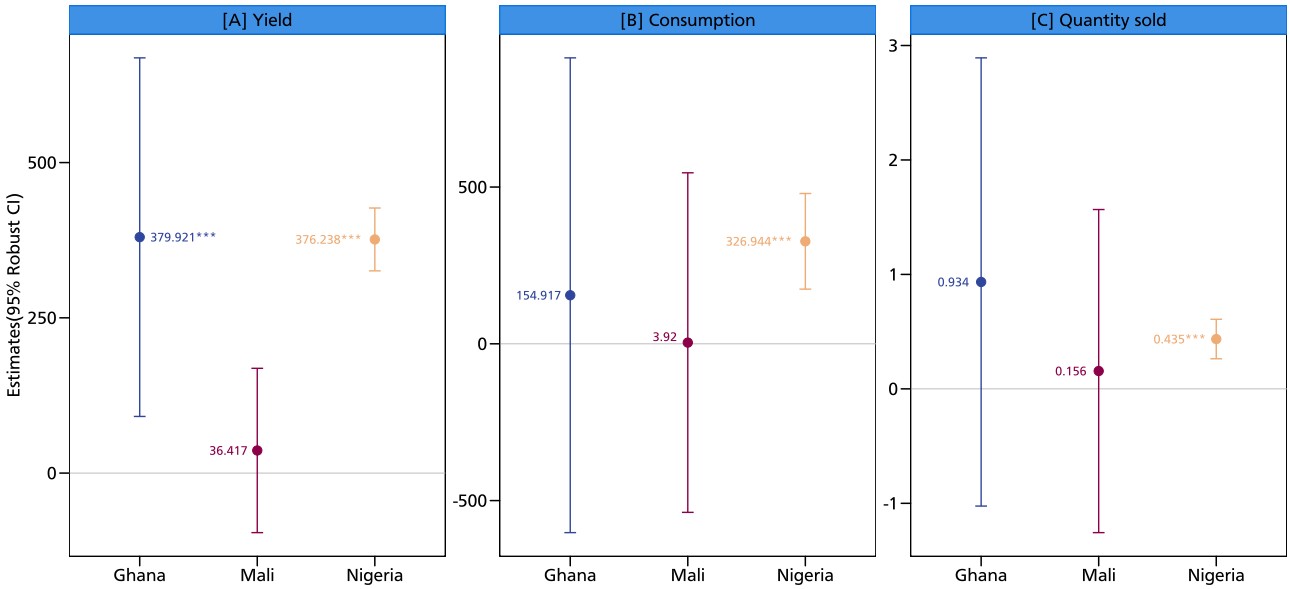

**Fig. 6 | Cross-country analysis of the relationship between adoption, production, consumption, and commercialization.** This figure displays coefficients along with their corresponding 95% confidence intervals as error bars. Panel **A** shows the cross-country heterogeneity impacts of climate-resilient groundnut varieties on yields, panel **B** shows the heterogeneity impacts on consumption per country, and panel **C** shows the impacts on quantity of groundnut sold per Ghana, Mali, and Nigeria. The coefficients are estimated using the two-stage least squares regression approach with $N = 8604$ observations. The presence of an asterisk (*) above a coefficient indicates that the coefficient is statistically different from zero at a predetermined level of significance (***$p < 0.01$, **$p < 0.05$, *$p < 0.1$). Statistical tests are two-sided $t$-tests. Full models are reported in Table S10 in the supplementary information. The models are estimated with additional controls such as age and education level of the household head, dependency ratio, gender of the household head, household size, cooperative membership, training, access to public and private extension, access to both cash and in-kind credit, distance to nearest urban and village market, crop rotation, mixed cropping, labor, market price, input costs, area of cultivation, off-farm income, and soil type. Source data are provided as a Source data file.

transformation. This analysis gives credence to the development, upscaling, and dissemination of various improved climate-resilient technologies, as they have the potential to boost smallholder commercialization. Harnessing the full gains from adoption of climate-resilient groundnut varieties may involve better management of seed systems and effective follow-up to ensure their sustained adoption. For example, it is important to ensure that seed delivery systems and markets are not missing and that transaction costs to access such

markets are minimal. Doing so could increase smallholder commercialization, with implications for welfare and rural development. While some of these recommendations do not directly emerge from this study, we raise them to place them in the larger context of the empirical literature[18,21,24]. The implications for building smallholders' resilience to climate change and extreme weather events are important. As such, climate-resilient crop varieties which constitute an important part of the CSA approach has the potential to offer some of

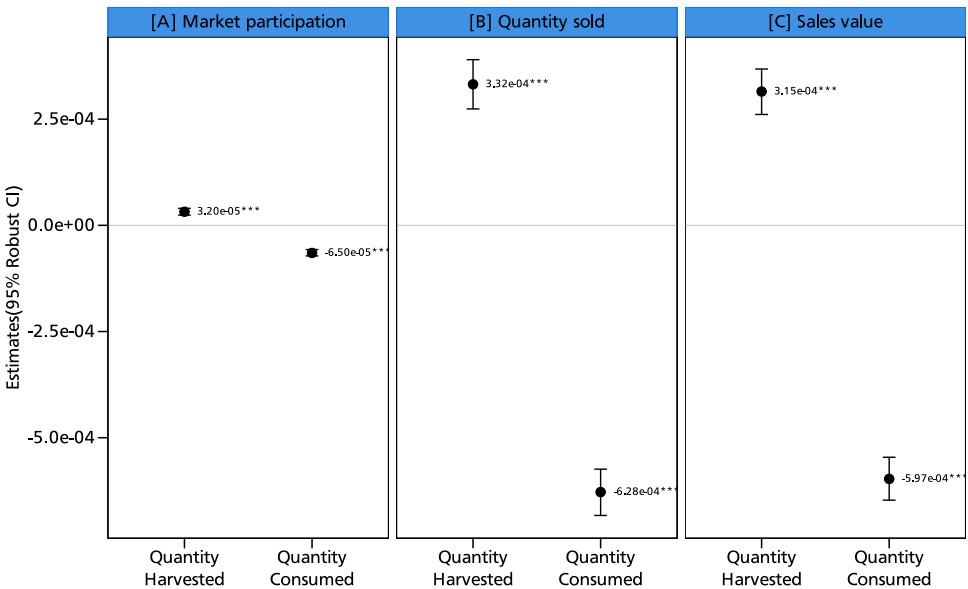

**Fig. 7 | Simultaneous effects of groundnut production and consumption on commercialization.** This figure displays coefficients along with their corresponding 95% confidence intervals as error bars. Panel **A** shows the production and consumption effects on market participation, panel **B** shows the production and consumption effects on quantity sold, and panel **C** shows the production and consumption effects on sales value. The coefficients are estimated using the two-stage least squares regression approach with $N = 8604$ observations. The presence of an asterisk (*) above a coefficient indicates that the coefficient is statistically different from zero at a predetermined level of significance (***$p < 0.01$, **$p < 0.05$, *$p < 0.1$). Statistical tests are two-sided t-tests. Full models are reported in Table S11 in the supplementary information. The models are estimated with additional controls such as age and education level of the household head, dependency ratio, gender of the household head, household size, cooperative membership, training, access to public and private extension, access to both cash and in-kind credit, distance to nearest urban and village market, crop rotation, mixed cropping, labor, market price, input costs, area of cultivation, off-farm income, and soil type. Source data are provided as a Source data file.

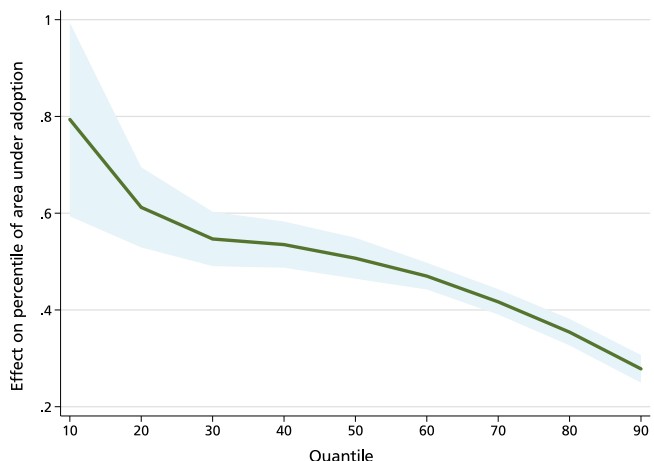

**Fig. 8 | Quantile estimates of groundnut adoption and commercialization.** This figure shows the result of a quantile regression between commercialization and adoption. $N = 8604$ observations. The green line shows the estimated impact of adoption per quantile. The gray area around the green line indicates the 95% confidence interval. Source data are provided as a Source data file.

the wins of CSA such as increasing productivity with ensuing implications on consumption and commercialization which constitute different aspects of smallholder welfare.

## Methods
### Ethics statement
This research complies with all relevant ethical regulations. The research proposal and data collection tools were approved by the ethics committee of the West and Central Africa Research Unit of ICRISAT.

### Survey design and data
This analysis is based on a three-wave panel dataset from three West African countries (Ghana, Mali, and Nigeria; see Fig. 9), where a farm household survey was conducted in 2017, 2018, and 2019. These countries were part of the United States Agency for International Development (USAID)-funded groundnut upscaling project implemented from 2015 to 2019. They were also part of the Feed the Future zone of influence and benefited recently from the activities of the project, which aimed to upscale groundnut productivity[25].

A multistage sampling procedure was used to select different regions and districts for data collection. The project targeted three regions in Ghana (Northern, Upper East, and Upper West) and Mali (Koulikoro, Mopti, and Sikasso) and five states in Nigeria (Jigawa, Katsina, Kano, Kebbi, and Sokoto). In these areas, the project selected some districts (in Ghana and Mali) and Local Government Areas (LGAs in Nigeria). In every selected district/LGA, some villages benefited from technology transfer activities implemented by the project, such as field schools, participatory demonstration plots, and innovation platforms. For the farm household survey, 4–6 villages were randomly selected from each district/LGA, from which about 30 households were further randomly selected. This sampling procedure put adopters and non-adopters under similar administrative, environmental, and climatic conditions. In the first year of data collection (2017), a total of 900 households from Ghana, 1,350 households from Mali, and 2500 households from Nigeria were surveyed.

During the second and third survey rounds, financial constraints reduced the initial sample by 30–40%. About 60–70% of households were randomly resampled from the initial sample. In Ghana, this translated into 540 households surveyed in 2018, from which 34 households dropped out in 2019 as a result of nonavailability during the survey period. Overall, a balanced sample of 498 households from Ghana was used in the analysis. In Mali, security conditions in the region of Mopti deteriorated in 2018 to the point that it was unsafe to conduct a survey. The 450 households initially surveyed in this region

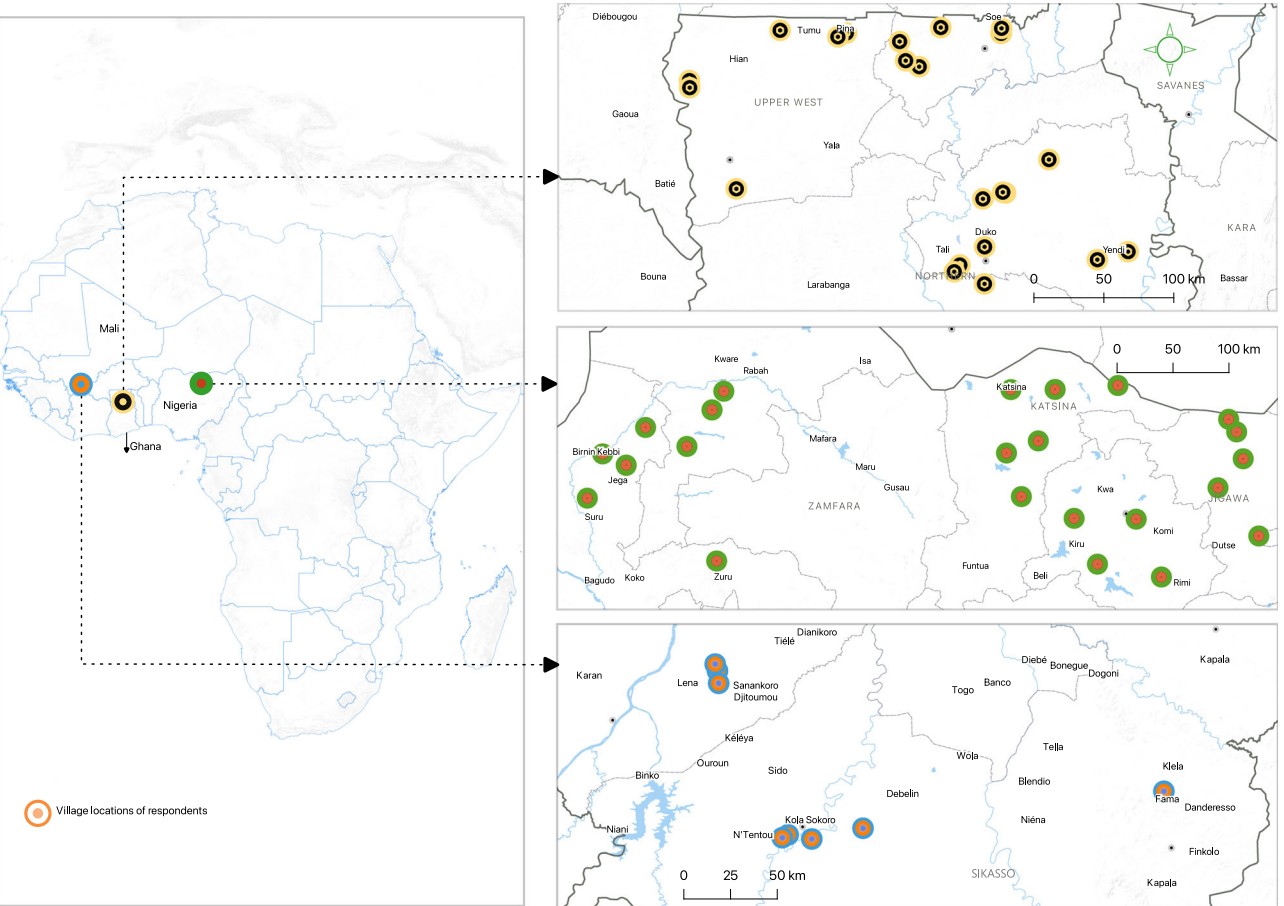

**Fig. 9 | Map of study countries.** This figure displays the study areas (Ghana, Mali and, Nigeria). The green and red colored circles represent survey villages in Nigeria; the yellow and black circles are for villages in Ghana and the blue-green and red- orange circles are for village sites in Mali. Households were randomly sampled in these villages.

were thus removed from the sample. The survey then proceeded with 900 households in the other two regions. Sixty households eventually dropped out in 2019 and a balanced sample of 840 households from Mali was used in the analysis. In Nigeria, 1600 households were randomly resampled in 2018 and 70 households dropped out in 2019, so the final analysis focused on a balanced sample of 1530 households. In total, we used 8604 observations collected from 2868 households over three years. Gender was not an integral part of the analysis as the focus was on households where we interviewed the household heads. So, there was no possibility of determining gender based on self-reporting or on assignment. The less focus on gender for the analysis was based on the expectation of impacts at the household as opposed to the individual level. Farming in many parts of West Africa is generally carried out by households, so we follow this reality to understand the impacts of the adoption of climate-resilient groundnut varieties on production, consumption, and commercialization.

Throughout the panel years, especially between 2018 and 2019, we recorded attrition rates of 8% in Ghana, 7% in Mali, and 4% in Nigeria. These attrition rates are considerably lower than those of other large household surveys in Africa. Probit models of attrition did not find evidence of bias in our estimates.

### Variable measurements
**Measurement of outcomes.** We measure commercialization using three different proxies. First, we use a binary measure of market participation that indicates whether households sell groundnuts in output markets. It takes the value of 1 for sellers and 0 otherwise. Second,

households that participate in markets sell at different intensities, so capturing households' actual sales levels may help to understand and differentiate sales intensity. Thus, we use the actual quantity of groundnut sold as the second proxy for commercialization. The value of groundnut sold, referred to as the sales value, is the third proxy for commercialization, measured using individual prices that farmers received for their output.

We also use three proxies to measure smallholder production: total quantity produced, production value, and yield. Like sales, we also value production. Yield is measured on a per hectare basis. Finally, we capture the quantity of groundnut that is consumed by households (kg).

**Measurement of adoption of climate-resilient groundnut varieties.** We measure adoption using two proxies: a binary variable that takes the value of 1 for adopters of climate-resilient groundnut varieties and 0 otherwise; and the area (hectares) under climate-resilient groundnut varieties. To avoid misidentification of the climate-resilient varieties, we focus on a list of varieties promoted in the area by USAID's Feed the Future program, the Gates Foundation-funded Tropical Legumes I, II, and III projects, and the recent USAID-funded groundnut upscaling project from 2015–2019. Some of these varieties include Samnut 22, Yenyawoso, and Nkatiesari in Ghana; ICGV 86124 (Niètatiga), ICGV 86015 (Yriwatiga), ICGV 86024 (Bonitiga), and Fleur 11 (Allason) in Mali; and Samnut 23, Samnut 24, Samnut 25, and Samnut 26 in Nigeria. Focusing on these varieties to identify adoption status should not lead to significant bias but rather to more precise measurements of the true adoption status. These climate-resilient groundnut varieties differ

from the landraces in these communities. They can withstand extended dry seasons (heat- and drought-resistant), diseases, and pests associated with groundnuts, such as the rosette virus[25,26]. They have been disseminated in the study areas for over a decade and are therefore well known, popular, and easily recognizable by farmers[13]. The enumerators were also trained to correctly capture the varieties farmers were using. We are cognizant of the literature on the misclassification of improved crop varieties, but since we focused on well-promoted and easily identifiable varieties, our analysis should not be prone to misclassification bias.

**Empirical estimation.** We are interested in understanding the relationship between adoption of climate-resilient groundnut varieties and production, consumption, and smallholder commercialization. Since we have panel data, we dive directly into panel data models for estimating these relationships, but before doing so, we pool the data and estimate the following regression equation:

$$Y_{it} = \alpha + A_i\delta + X_i\beta + \mu_i \qquad (1)$$

$Y_{it}$ represents the various outcomes of interest, including production (harvested quantity, production value, and yield), consumption, and commercialization (market participation, quantity sold, and sales value). Our variable of interest is $A_i$; its parameter estimate $\delta$ shows the relationship between adoption and the various outcomes. We estimate different models for the adoption dummy and the extent of adoption (area under adoption). $X_i$ is a vector of both farm- and household-level control variables; $\mu_i$ is the stochastic error term. Equation (1) can be estimated using the naive ordinary least square (OLS) estimator. Linear models are always preferred for causal identification as they are easy to interpret and do not lead to identification by functional form, which is common with some maximum likelihood procedures. However, the results may be biased by both observed and time-invariant unobserved heterogeneity. Notwithstanding, the OLS regression outputs are reported in the supplementary tables (S2–S5). As we have panel data, we exploit its quality to address any time-invariant unobserved heterogeneity. We employ IV estimators to control for time-varying observed factors and test for robustness with the control function (CF) approach.

We rely on standard two-stage least squares (2SLS) regressions to analyze how production, consumption, and commercialization are affected by adoption of climate-resilient groundnut varieties. The second stage of the 2SLS model is represented as:

$$Y_{it} = \alpha + A_{it}\delta + X_{it}\beta + d_t + c_{1i} + \mu_{it} \qquad (2)$$

As shown in Eq. 2, we now introduce the time component and fully explore the panel data. The same symbols are used, but additionally $c_{1i}$ represents time-invariant unobserved heterogeneity, $d_t$ is time-fixed effects, and $\mu_{it}$ is the stochastic error term. Our panel data enable us to effectively control for time-invariant unobserved heterogeneity like skills, preferences, and motivation, which may drive both adoption and the outcomes. Two common estimators are the household FE and random effect (RE) estimators. The choice of any estimator depends on the assumptions about correlations between unobserved heterogeneity and the observed characteristics. It may also depend on the level of within variation in the outcomes and the length of the panel. For linear models, the FE estimator has been used as the workhorse in controlling time-invariant unobserved heterogeneity. However, this estimator could lead to the incidental parameters problem for nonlinear models. The RE estimator, on the other hand, is quite restrictive and is more commonly used in experimental studies since it assumes strict exogeneity (no correlation) between the observed covariates and unobserved heterogeneity. To relax this strict assumption, the Mundlak-Chamberlain device, also known as the CRE model, is

recommended. This model assumes that this correlation is a linear function of the average across time of all time-variant covariates in Eq. (2)[27]. This estimator has several advantages over both the FE and RE estimators: (1) it relaxes the strict exogeneity assumption of the RE estimator; (2) it provides more efficient estimates than the FE estimator when the within variation in data is smaller than the between variation; and (3) it avoids the incidental parameters problem for nonlinear models. We use the CRE model and test robustness by also specifying the FE estimator. In terms of application, the CRE model is similar to the RE model but with the addition of time averages of all time-varying covariates ($\tilde{X}_i$), as shown in Eq. (3):

$$Y_{it} = \alpha + A_{it}\delta + X_{it}\beta + \tilde{X}_i\gamma + d_t + c_{1i} + \mu_{it} \qquad (3)$$

Now that we have addressed unobserved heterogeneity, we are still left with two other sources of endogeneity: reverse causality and measurement error. One could argue that there exist reverse causality concerns between adoption and production, consumption, and commercialization. While adoption of climate-resilient groundnut varieties could lead to commercialization through higher yields, commercialization could also result in higher adoption if the gains from commercialization are used to purchase the climate-resilient seeds. It is likely that adoption, therefore, correlates with time-varying shocks. Regarding measurement error, it is always challenging to claim the accuracy of the data-generating process. However, we are certain that adoption was properly captured as explained above, since these processes were well supervised and monitored. To reduce endogeneity concerns associated with reverse causality and measurement error, we employ the IV approach, as represented in the first stage of the 2SLS model (Eq. 4).

$$A_{it} = Z_{it}\delta + X_{it}\beta + d_t + c_{2i} + \epsilon_{it} \qquad (4)$$

where $Z_{it}$ refers to the IV. As highlighted by Angrist and Pischke[28], the use of an IV also helps in correcting any biases from measurement errors. Selecting instruments is not a trivial process as they must be exogenous and satisfy the exclusion restriction. Good instruments should normally involve some form of randomization to be able to induce an exogenous variation for causal claims.

We use willingness to adopt climate-resilient groundnut varieties as the instrument, as it can be argued to envelop subjective preferences for adopting such varieties. This variable is potentially correlated with both observed and unobserved characteristics such as skills, preferences, and managerial abilities. Previous studies have shown that the use of willingness to pay variables can help control for any residual endogeneity[29,30]. Willingness to adopt climate-resilient crop varieties is defined as a dummy variable that takes the value of 1 for households that have access and are willing to adopt climate-resilient groundnut varieties, and 0 otherwise. Accessibility in this case implies some aspects of awareness (knowledge exposure) of climate-resilient groundnut varieties. Of course, households will only adopt them if they know about, have access to, and are willing to adopt them. Information exposure usually matters and has been shown to drive adoption of climate-resilient groundnut varieties, sometimes coupled with knowledge exposure[31]. Beyond being aware and having access, liquidity matters and has been argued to be an important constraint in technology adoption, since households will only adopt if they are not cash-strapped[2].

Our IV meets the criteria required for it to be relevant as it exhibits a strong partial correlation with adoption of climate-resilient groundnut varieties. Estimating Eq. 4 shows that willingness to adopt climate-resilient groundnut varieties is significantly associated with their adoption ($p < 0.000$), and the F-statistic is 357.5, which is above the threshold value for weak instruments[32]. Regarding instrument exogeneity, we maintain exogeneity as the IV is likely not correlated with

the household-level time-varying errors, especially since we have controlled for observed covariation and time-invariant unobserved heterogeneity. Of course, the use of the different controls eliminates potential channels through which the exclusion restriction may be violated. That said, there are usually no valid tests for exclusion restriction and our instrument may not be perfect. Still, we present a battery of robustness checks on the identification strategy, particularly the IV estimation. As part of this, we employ the Hausman Taylor IV estimator, which estimates time-invariant covariates[33,34]. We also employ the[35] heteroskedasticity-based estimator that generates internal instruments by exploiting heteroskedastic covariance restriction in the presence of weak or no instruments. Finally, we use the two-stage residual inclusion approach[36]. All these different specifications point to the robustness of our estimation, as we obtain similar effects for all these different estimators.

Beyond the average treatment effect and local average treatment effect obtained from the OLS and IV specifications, we perform some quantile regressions to understand the association between adoption and different quantiles of the conditional distribution of commercialization.

**Sustained adoption over time.** We next seek to understand the role of sustained adoption given the seeming adoption-disadoption that is common with the adoption of improved crop varieties. To do so, we generate a sustained adoption variable representing adoption over the three panel years. We verify whether those who continuously used climate-resilient groundnut varieties for all three years obtained higher yields, consumed more groundnuts, and realized greater commercialization gains than households that only adopted them for one or two years. The construction of this variable is akin to the treatment and control groups generated based on the continuous adoption of improved chickpea varieties[37].

**Robustness checks.** We perform several robustness tests to confirm and strengthen the findings. First, as an alternative identification strategy, we employ the two-stage residual inclusion (2SRI) approach, which usually leads to the 2SLS in linear models, especially when the endogenous independent variables are linear in parameters. The 2SRI approach, also known as the CF, provides a direct test for endogeneity. Besides being easy to compute, it requires less restrictive assumptions than maximum likelihood estimation techniques[36]. It addresses endogeneity by including the residuals of the endogenous variable obtained in the first-stage model in the second-stage model, in the place of predicted probabilities. In doing so, it assumes the normality of the second-stage model conditional on the endogenous variable and the residual from the first-stage model. One particular caveat in using the 2SRI is that to obtain consistent estimates, the same set of explanatory variables except for the IVs should be used in the first- and second-stage regressions. The 2SRI approach involves running an adoption model in the first stage on other controls with the addition of instruments. In the second stage, the generalized residual obtained in the first stage is modeled together with the outcomes and other controls. As shown in Table S12 in the supplementary information, we obtain results similar to those of the 2SLS approach.

Alternatively, we employ the Hausman-Taylor Instrumental Variable (HTIV) estimator to correct for endogeneity and make the main findings more robust. Akin to the IV approach, this estimator does not rely on external instruments but rather finds instruments within the model[33,34]. It exploits both between and within variations of the exogenous variables and uses them as instruments. Being a panel estimator, it has an edge over the FE estimator since it provides estimates for time-invariant variables. In addition, it can correct for endogeneity arising from simultaneity and reverse causality. We estimate our main commercialization equations as

shown in Table S13 in the supplementary information. The findings here are again consistent with the IV estimates, both in magnitude and statistical significance, bolstering the main findings that technology adoption is associated with smallholder commercialization.

Finally, we use the Lewbel's IV approach[35], a heteroskedasticity-based estimator that generates internal instruments exploiting heteroskedastic covariance restrictions in the traditional structural model sense. This method is generally used to test the validity of instruments in regression settings[35,38,39]. It has the advantage of testing over-identifying restrictions, as it specifies the Hansen J statistic and the Hayashi C test of excluded instruments validity. We estimate two sets of models: one where we allow the model to generate internal instruments, and one where we augment the use of the IV with the internally constructed instruments. We again find numerically similar estimates for the two sets of models and the commercialization outcomes, as shown in Table S14 in the supplementary information. These results further support and strengthen our main findings, increasing estimation efficiency. All these checks provide reassuringly similar results despite using various estimators with different identifying assumptions.

Descriptive statistics and econometrics analysis were performed in STATA 17 and R 4.3.1 was additionally used to generate the figures.

### Reporting summary

Further information on research design is available in the Nature Portfolio Reporting Summary linked to this article.

### Data availability
The data used in this study have been deposited in the zenodo repository[40]. Source data are available at this repository[40]. Source data are provided with this paper.

### Code availability
Code for replication of the results as well as figures in the manuscript are available at the zenodo repository[40].

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

## Acknowledgements

We thank the United States Agency for International Development (USAID) and the International Crops Research Institute for the Semi-Arid Tropics (ICRISAT) for the data used in the study. The data were collected as part of the USAID-funded project "Increasing Groundnut Productivity of Smallholder farmers in Ghana, Mali, and Nigeria" implemented by ICRISAT. This work also benefited from the support of the CGIAR research initiative on National Policies and Strategies (NPS). Additional support from the German Research Foundation under CRC/Transregio 228: Future Rural Africa: Future-making and social-ecological transformation (Project number: 328966760) is also acknowledged. All remaining errors are ours. The usual disclaimer applies.

## Author contributions

M.P.Jr.T.O: Conceptualization, methodology, data analysis, original draft preparation, writing, and editing. J.C.L.: Conceptualization, data collection, sampling design, methodology, data analysis, original draft preparation, writing, and editing. B.H.G.: Conceptualization, Methodology, data analysis, visualization, original draft preparation, writing, and editing. H.D.A.: Data collection, sampling design, visualization, writing, and editing.

## Competing interests

The authors declare no competing interests.
