## [Peer Review File · Nature Communications]

Reviewers' Comments:

Reviewer #1:

Remarks to the Author:

This paper examines the relations between improved climate-smart crop varieties, land productivity, and smallholder commercialization using evidence from three West African countries. While each of these areas are important and topical for literature on smallholder agriculture, exploring the links between them makes for a unique and great contribution. The results are interesting for both academics and development practitioners. The research methodology is rigorous and the dataset employed is rich and credible. The econometric models and dataset are clearly presented in a way that allows for the work to be reproduced. By employing two-stage residual inclusion to check for robustness the paper confirms the strength of the findings. I have not detected any flaws in data analysis and interpretation.

I only have one minor suggestions: Given the strength of the evidence from this paper, perhaps the authors could expand the concluding section by discussing policy implications. This is especially important given the significant public and private sector investments in smallholder agriculture (and specifically groundnut value chains) in West Africa. Expanding this section would help to answer a 'so what?' question for some readers that might be less familiar with smallholder agriculture in sub-Saharan Africa.

Reviewer #2:

Remarks to the Author:

The manuscript deploys many different estimation strategies to explore the implications of modern groundnut variety adoption.

Overall, the manuscript seems to focus too much on using many econometric approaches without paying sufficient attention to how they jointly contribute to the exploration of the problem and leave the reader to do most of the interpretation. The manuscript lacks detail in several aspects and overstates its contributions and results in many places. Furthermore, a lack of attention to detail in language, terminology and references adds to the overall impression that it is not yet ready for publication. Despite the problems in the current version of the manuscript, the analysis and data appear to offer interesting and new insights that could be valuable. With a little more thought on the interpretation and a clearer focus on what the data represent this could turn into a good paper.

Results:

- Figure 4 shows consistently negative effects for area under adoption. In subsequent figures the trend that the effects are lower or even negative is consistent, yet this is not mentioned or explained in any way. Instead the positive effect for the generic adoption is highlighted and used as a basis for the conclusions.
- Given the highly significant differences across the 3 countries as shown in Figure 10, it would be important to reflect on the differences between the countries and the potential influence this may have for the results that are presented. Some insights can be drawn from the supplementary material, yet this is left to the reader to explore.
- The decomposition of direct and indirect effects remains completely vague. It is not further elaborated what the direct effect could be while it is asserted that the indirect effect is on yield. In fact, the most direct effect of modern varieties is the yield increase which leaves the reader wonder what the rationale is behind the classification as the indirect effect.
- Relatedly, the effects outlined in the manuscript are for an adoption cut off at 3 years. In the supplementary material (page 43) an alternative with 2 years is presented as a robustness check. There the indirect effect is no longer significant while it is highly significant in the results presented in the manuscript. What would be the reasons for this and why is a robustness check performed when the results appear not robust and there is no discussion of this result.
- Line 230: "This suggests that sustained adoption is more effective in increasing market participation, quantity sold and sales value." It seems more likely a case of self-selection of the successful farmers into expanding the production of new varieties. Certainly it would need to be substantiated in the analysis.

- Line 262: "In sum, it turns out that the relationship between adoption and commercialization can be explained entirely by increases in land productivity." This is only true under the model specification in the manuscript (not in the one used as a robustness check) and only if the indirect effect is indeed interpreted as land productivity.
- Line 264: "This suggests that commercialization would increase more if land productivity increases" This is highly convoluted statement and does not emerge from the analysis as stated.
- Section 5.3 attempts to decompose the results by farm size. There are several problems here. The statement that small farms are more productive (line 281) is substantiated by the finding that they benefit more from commercialization. However, commercialization is not the same as productivity. This misinterpretation is the only basis for the claim that smaller farms are more productive.
- In the same paragraph it is stated that the small farm productivity finding is often based on measurement error. It is then stated that "taking away this measurement error, ..." (line 284). There is no information on how or in fact if the measurement error was reduced or removed.

Data:

- Line 352 states that half beneficiaries were selected. Assuming what is meant that half of the sampled households are beneficiaries, the open question remains how the beneficiaries were selected or targeted by the project. Was the seed offered to everyone and some self-selected in or was there a targeting mechanism that may influence the results?

Conclusions:

- Line 576: "This result is suggestive of the fact that specialization which may be proxied for by the scale of adoption is positively associated with commercialization" - The groundnut area under the new varieties is not a proxy for specialization on its own. Any indication of specialization can not take the groundnut area on its own but has to look at the whole farm operations as well as any other income streams such as business incomes. As the manuscript doesn't include this, this conclusion is not supported.
- Line 585: "... various improved climate smart technologies as they have the potential to unlock market opportunities." - Besides the mentioned problem that this generalization to all crops is not supported, the manuscript does not analyze how this suggested unlocking of market opportunities is happening. It does suggest that farmers are increasing their marketed share but unlocking markets implies that these were previously not accessible which is not the case. The subsequent sentences are adding further concepts that were not part of the analysis. Seed systems, seed delivery or transaction costs have not been mentioned before and have not been part of the analysis. Recommendation in this area are unfounded.

General points:

1. Terminology is not used with care. Two important and indicative examples are:

- It is often referred to "climate smart crops" in general while the paper only analyzed groundnuts and results cannot be generalized to crops in general. In addition, there is little to no evidence that the groundnut varieties under investigation are climate smart. There is no definition or analysis on what could make them climate smart either. I would argue these are traditional yield enhancing varieties that have few if any climate smart attributes.
- The manuscript uses land productivity when it talks about yields (starting from line 116 "land productivity (yields)"). Land productivity is a different measure and concept. What appears to be used here is yield which is appropriate.

2. There are some structural issues that need to be addressed:

- The last 3 paragraphs of the introduction are far too detailed and provide more than a preview of the results. They are not well structured and do not facilitate a higher level overview of what will be presented later.
- The results and discussion section is placed directly after the introduction and data and methods are only provided after the results are presented. This is highly unusual and leave many questions open while reading the results.

3. The language needs close attention as many sentences are confusing or misleading or just not appropriate – some might be typos. Examples below:

- Line 41 "The Green revolution which recorded numerous strides in increasing agricultural production ..." - besides the odd phrasing, it might be appropriate to also mention the negative effects of the Green Revolution which are well documented.
- Line 69: "While this is not much of a problem given that multi-period data are not easy to come by ..." - it remains unclear if the single period data is considered problematic or not given subsequent statements. That something is "not easy to come by" is not a satisfactory basis to judge the appropriateness of a data source.
- Line 116: "... land productivity (yields) and other higher other outcomes like commercialization." - likely other should be order
- Line 130: "Moreover, it adds some intuitive insights into the traditional claim that "small is beautiful"." - The claim here is not well founded and the alleged traditional claim should either be referenced or further elaborated
- Line 164: "because of the green revolution which led to the development of high-yielding varieties" - In the traditional reading, it is rather the other way around in that the development of high yielding varieties (and corresponding agronomic advice) led to the green revolution taking place.
- Line 211: "When we consider the extent of adoption, the results are still maintained though the magnitudes reduce in size to somewhat a lower bound." - Statements like this need further explanation. It is not self-evident that this would constitute the lower bound
- Line 398: "We are cognizant of the literature on the misclassification of improved crop varieties but given that we focused on well-promoted and easily identifiable varieties, we do not feel our analysis is pruned to misclassification bias." - The word feeling is inappropriate here and given the potentially large influence this may have it would need a little more elaboration of the issue.

4. References are not always accurate and sometimes misrepresent the content. Several references are used in a way that implies they treated groundnut variety adoption or outcomes. However, they analyzed other crops in other countries. Some results but more likely the methods could be applicable, however. Some examples are:

- Verkaart et al 2017 – The paper is about Chickpea in Ethiopia
- Simtowe et al 2019 – The paper is about Maize in Uganda
- Tabe-Ojong et al 2022b – The paper is about Chickpea in Ethiopia

Response to Reviewer's comments

The Adoption of Climate-Resilient Groundnut Varieties Increases Agricultural Production and Smallholder Commercialization in West Africa

Dear Reviewers,

Thank you for your constructive comments and suggestions. Your comments have helped us to improve the overall quality of the paper. Following your suggestions, we have made substantial revisions which we believe have improved the readability of the paper. Below, we reproduce your comments in plain text and beneath it, we provide our replies in blue text. We hope that our revisions have fully addressed your comments.

Reviewer #1 (Remarks to the Author):

This paper examines the relations between improved climate-smart crop varieties, land productivity, and smallholder commercialization using evidence from three West African countries. While each of these areas are important and topical for literature on smallholder agriculture, exploring the links between them makes for a unique and great contribution. The results are interesting for both academics and development practitioners. The research methodology is rigorous and the dataset employed is rich and credible. The econometric models and dataset are clearly presented in a way that allows for the work to be reproduced. By employing two-stage residual inclusion to check for robustness the paper confirms the strength of the findings. I have not detected any flaws in data analysis and interpretation.

We thank the reviewer for the thorough evaluation of our manuscript and for highlighting its strengths and significance. We are pleased that you see high value in our analysis and specify how to better sell the results of the manuscript by critically discussing the policy implications

I only have one minor suggestions: Given the strength of the evidence from this paper, perhaps the authors could expand the concluding section by discussing policy implications. This is especially important given the significant public and private sector investments in smallholder agriculture (and specifically groundnut value chains) in West Africa. Expanding this section would help to answer a 'so what?' question for some readers that might be less familiar with smallholder agriculture in sub-Saharan Africa.

Thank you for this comment which we have followed. We have now added some policy implications to the ones we originally had. The section now reads better and has some nice implications on how to upscale groundnut value chains in West Africa especially linking it to climate change.

Reviewer #2 (Remarks to the Author):

The manuscript deploys many different estimation strategies to explore the implications of modern groundnut variety adoption. Overall, the manuscript seems to focus too much on using many econometric approaches without paying sufficient attention to how they jointly contribute to the exploration of the problem and leave the reader to do most of the interpretation. The manuscript lacks detail in several aspects and overstates its contributions and results in many places. Furthermore, a lack of attention to detail in language, terminology, and references adds to the overall impression that it is not yet ready for publication. Despite the problems in the current version of the manuscript, the analysis and data appear to offer interesting and new insights that could be valuable. With a little more thought on the interpretation and a clearer focus on what the data represent this could turn into a good paper.

We thank the reviewer for the constructive comments, recommendations, and suggestions to make the paper better. We apologize for the lack of attention to detail in language, and terminology and for being slopping with our references. We have followed all your recommendations and suggestions in improving the manuscript. In the first place, we have removed some of the econometric models as they were just robustness checks and we already showed that our results and findings are robust to different estimation strategies and assumptions. Specifically, we have removed the baseline OLS estimation and limited the analyses to three econometric models: 2SLS to estimate the causal effect of adoption on land production and commercialization, quantile regression to estimate heterogeneous effects, and mediation analysis to decompose the total commercialization effect into direct and indirect effects. In addition, we have provided more details and improved interpretation where necessary even beyond your recommendations. Similarly, we moderated our statements about the contribution and results of the study. The manuscript has also benefited from language editing by a professional English editor. We believe our manuscript is now very much improved and has benefitted from your kind and detailed review.

Results:

- Figure 4 shows consistently negative effects for area under adoption. In subsequent figures the trend that the effects are lower or even negative is consistent, yet this is not mentioned or explained in any way. Instead the positive effect for the generic adoption is highlighted and used as a basis for the conclusions.

You are right that the effects for area under adoption are negative but as you'll observe from the same figure, these effects are not statistically significant even at the 10 percent level of probability. It is for this reason that we do not interpret them or mention them anywhere in the manuscript as they can not be used for any broad generalization. We instead explain and interpret the generic adoption as it is statistically significant at different levels of probability. That notwithstanding, we drop the baseline OLS model as they are biased and do not account for potential endogeneity concerns of adoption. These results can be regarded as a benchmark, reason we now leave them in the supplementary material.

- Given the highly significant differences across the 3 countries as shown in Figure 10, it would be important to reflect on the differences between the countries and the potential influence this may have for the results that are presented. Some insights can be drawn from the supplementary material, yet this is left to the reader to explore.

Thank you for this comment. We have now elaborated on the results of the cross-country analyses by providing contextual insights that may help to understand the observed differences between countries. Some of these differences relate to available market infrastructure and the price incentive structure as well as the household structure and characteristics which could significantly explain consumption patterns. This relationship is in line with insights from the non-separable agricultural household model where households would only approach markets as sellers to the extent that their household food demands are met (Tabe-Ojong et al.,2022).

- The decomposition of direct and indirect effects remains completely vague. It is not further elaborated what the direct effect could be while it is asserted that the indirect effect is on yield. In fact, the most direct effect of modern varieties is the yield increase which leaves the reader wonder what the rationale is behind the classification as the indirect effect.

Thank you for this comment. It appears there was some confusion here. You are right that the most common direct effect of modern varietal adoption is yields but in the context of the mediation analysis that we are doing, it is an indirect effect as we have throughout assumed that the direct effect is on commercialization. All this information is presented in section 4.3 (Mediation analyses) under section 4 (Methodology). There, we specified that the direct effect is the effect on commercialization while the indirect effect is the effect that runs through yields, which is the mediator that we are testing under this section.

- Relatedly, the effects outlined in the manuscript are for an adoption cut off at 3 years. In the supplementary material (page 43) an alternative with 2 years is presented as a robustness check. There the indirect effect is no longer significant while it is highly significant in the results presented in the manuscript. What would be the reasons for this and why is a robustness check performed when the results appear not robust and there is no discussion of this result.

There appears to still be some confusion here and some wrong inferences. In Tables 14 and 15 in the supplementary material, we presented the results of continuous adoption (3years) which we robustify with adoption only for two years (Tables 22 and 23). For both of these regressions, the indirect effects are not statistically significant and just the direct effect on commercialization is significant. These regressions are for testing sustained (continuous) adoption and they are quite different from the normal estimations where we estimate adoption at any of the panel years. Here, both the total, direct, and indirect effects are statistically significant for quantity sold and sales value (Tables 12 and 13). We have however decided to remove the robustness checks for these regressions, including the sustained regressions as they do not add any particular value to the manuscript and like you highlight may not be relevant.

- Line 230: “This suggests that sustained adoption is more effective in increasing market participation, quantity sold and sales value.” It seems more likely a case of self-selection of the successful farmers into expanding the production of new varieties. Certainly it would need to be substantiated in the analysis.

We do not think self-selection has a role to play here as we exploited the panel nature of our data to control for self-selection using both the household fixed effect estimator and the correlated random effect estimator. Beyond this, we also used instrumental variable estimators to reduce selection bias. We detailly discuss all these under section 4 (methodology).

- Line 262: “In sum, it turns out that the relationship between adoption and commercialization can be explained entirely by increases in land productivity.” This is only true under the model specification in the manuscript (not in the one used as a robustness check) and only if the indirect effect is indeed interpreted as land productivity.

Thank you for this and you are correct that this relationship only holds when we consider adoption on a single or disjointed basis. We have dropped the mediation analysis for sustained adoption in this version of the manuscript.

- Line 264: “This suggests that commercialization would increase more if land productivity increases” This is highly convoluted statement and does not emerge from the analysis as stated.

We agree with your evaluation, and we have removed the statement.

- Section 5.3 attempts to decompose the results by farm size. There are several problems here. The statement that small farms are more productive (line 281) is substantiated by the finding that they benefit more from commercialization. However, commercialization is not the same as productivity. This misinterpretation is the only basis for the claim that smaller farms are more productive.

Thank you for this comment, we have now taken out some of these sentences as they do not represent our analysis.

- In the same paragraph it is stated that the small farm productivity finding is often based on measurement error. It is then stated that ‘taking away this measurement error, ...’ (line 284). There is no information on how or in fact if the measurement error was reduced or removed.

We have also taken out this statement.

Data:

- Line 352 states that half beneficiaries were selected. Assuming what is meant that half of the sampled households are beneficiaries, the open question remains how the beneficiaries were

selected or targeted by the project. Was the seed offered to everyone and some self-selected in or was there a targeting mechanism that may influence the results?

This was a mistake, which we have now corrected. Access to the seeds was really the decision of households as they had to adopt them after some open promotional activities. This brings in some aspects of self-selection which as we mentioned above, we controlled through the use of our panel data and various household fixed effects and instrumental variable regressions.

Conclusions:

- Line 576: “This result is suggestive of the fact that specialization which may be proxied for by the scale of adoption is positively associated with commercialization” - The groundnut area under the new varieties is not a proxy for specialization on its own. Any indication of specialization can not take the groundnut area on its own but has to look at the whole farm operations as well as any other income streams such as business incomes. As the manuscript doesn't include this, this conclusion is not supported.

Thank you for this comment which we agree with. We have now taken this statement from the conclusion.

- Line 585: “... various improved climate smart technologies as they have the potential to unlock market opportunities.” - Besides the mentioned problem that this generalization to all crops is not supported, the manuscript does not analyze how this suggested unlocking of market opportunities is happening. It does suggest that farmers are increasing their marketed share but unlocking markets implies that these were previously not accessible which is not the case. The subsequent sentences are adding further concepts that were not part of the analysis. Seed systems, seed delivery or transaction costs have not been mentioned before and have not been part of the analysis. Recommendation in this area are unfounded.

We have taken out most of these recommendations and are now more focused in the recommendations from the study. Most of the recommendations on seed systems and transaction costs are not unfounded as a conceptual framework linking adoption to commercialization shows that adoption can increase commercialization through two pathways (Tabe-Ojong et al.,2022). Adoption of improved varieties can reduce fixed transaction costs associated with participating in markets (Key et al. 2000; Tabe-Ojong et al. 2022). Fixed transaction costs refer to the 3 main cost distinctions: (1) search costs associated with searching for the market or customer with the best prices, (2) negotiation and bargaining costs which is related to search cost but common in situations of information asymmetry about prices, especially where households have to effectively bargain to get the best prices for the outputs and (3) sorting and time implication cost which refers to the time spent in making the output acceptable for market conditions. Beyond reducing fixed transaction costs and ensuring smallholder commercialization, the adoption of improved seeds may lead to farm production and productivity increases (Vaiknoras and Larochele 2021).

General points:

1. Terminology is not used with care. Two important and indicative examples are:

- It is often referred to “climate smart crops” in general while the paper only analyzed groundnuts and results cannot be generalized to crops in general. In addition, there is little to no evidence that the groundnut varieties under investigation are climate smart. There is no definition or analysis on what could make them climate smart either. I would argue these are traditional yield enhancing varieties that have few if any climate smart attributes.

Thank you for this comment which we have followed. In the first place, we consistently replace “climate-smart crops” with “climate-resilient groundnut varieties”. Second, these varieties are climate resilient as they are drought resistant and able to withstand the arid and semi-arid temperatures common in the study countries. As a matter of fact, they were bred for this purpose; to be climate resilient (heat and drought resilient). In this version of the manuscript, we provide some of the climate-smart attributes of the groundnut varieties under analysis (Acevedo et al., 2020).

- The manuscript uses land productivity when it talks about yields (starting from line 116 “land productivity (yields)”). Land productivity is a different measure and concept. What appears to be used here is yield which is appropriate.

Thank you for this comment, which we have followed. We have used yields throughout the paper.

2. There are some structural issues that need to be addressed:

- The last 3 paragraphs of the introduction are far too detailed and provide more than a preview of the results. They are not well structured and do not facilitate a higher level overview of what will be presented later.

Thank you for this great comment. We have taken out some of these statements to reduce the ease in following and understanding the overall manuscript.

- The results and discussion section is placed directly after the introduction and data and methods are only provided after the results are presented. This is highly unusual and leave many questions open while reading the results.

We agree that this is an unusual way to proceed, but it is to comply with the journal's standards and instructions. Nature journals require you to have and discuss the results before going to the data and methods section.

3. The language needs close attention as many sentences are confusing or misleading or just not appropriate – some might be typos. Examples below:

The revised version of the manuscript benefited from language editing. We have also been keen to remove all typos and perform some editorial changes.

- Line 41 “The Green revolution which recorded numerous strides in increasing agricultural production ...“ - besides the odd phrasing, it might be appropriate to also mention the negative effects of the Green Revolution which are well documented.

We have added some of the negative effects as a footnote.

- Line 69: “While this is not much of a problem given that multi-period data are not easy to come by ...” - it remains unclear if the single period data is considered problematic or not given subsequent statements. That something is “not easy to come by” is not a satisfactory basis to judge the appropriateness of a data source.

We have taken out the whole paragraph as it does not add any specific value to the manuscript.

- Line 116: “... land productivity (yields) and other higher other outcomes like commercialization.“ - likely other should be order

Thank you for this, we have corrected.

- Line 130: “Moreover, it adds some intuitive insights into the traditional claim that “small is beautiful”.” - The claim here is not well founded and the alleged traditional claim should either be referenced or further elaborated

We have taken out this statement.

- Line 164: “because of the green revolution which led to the development of high-yielding varieties” - In the traditional reading, it is rather the other way around in that the development of high yielding varieties (and corresponding agronomic advice) led to the green revolution taking place.

We have rephrased the sentence.

- Line 211: “When we consider the extent of adoption, the results are still maintained though the magnitudes reduce in size to somewhat a lower bound.” - Statements like this need further explanation. It is not self-evident that this would constitute the lower bound

We have now added an explanation for this in the current version of the manuscript.

- Line 398: “We are cognizant of the literature on the misclassification of improved crop varieties but given that we focused on well-promoted and easily identifiable varieties, we do not feel our analysis is pruned to misclassification bias.” - The word feeling is inappropriate here and given the potentially large influence this may have it would need a little more elaboration of the issue.

We have corrected this and elaborated more on this issue as you'll see in this version of the manuscript.

4. References are not always accurate and sometimes misrepresent the content. Several references are used in a way that implies they treated groundnut variety adoption or outcomes. However, they analyzed other crops in other countries. Some results but more likely the methods could be applicable, however. Some examples are:

- Verkaart et al 2017 – The paper is about Chickpea in Ethiopia
- Simtowe et al 2019 – The paper is about Maize in Uganda
- Tabe-Ojong et al 2022b – The paper is about Chickpea in Ethiopia

Thank you for this comment, we have tried to reduce these inconsistencies, but as you notice, these studies were cited under a general context when we consider groundnuts as a legume alongside chickpea or drought resistant varieties like the maize case in Uganda.

References

Acevedo, M., Pixley, K., Zinyengere, N., Meng, S., Tufan, H., Cichy, K., Bizikova, L., Isaacs, K., Ghezzi-Kopel, K., & Porciello, J. (2020). A scoping review of adoption of climate-resilient crops by small-scale producers in low- and middle-income countries. *Nature Plants*, 6(10), Article 10. <https://doi.org/10.1038/s41477-020-00783-z>

Key, Nigel; Sadoulet, Elisabeth; Janvry, Alain de (2000): Transactions Costs and Agricultural Household Supply Response. In *American Journal of Agricultural Economics* 82 (2), pp. 245–259. DOI: 10.1111/0002-9092.00022.

Tabe-Ojong, Martin Paul; Mausch, Kai; Woldeyohanes, Tesfaye B.; Heckeley, Thomas (2022b): Three hurdles towards commercialisation: integrating subsistence chickpea producers in the market economy. In *European Review of Agricultural Economics* 49 (3), pp. 668–695. DOI: 10.1093/erae/jbab023.

Vaiknoras, Kate; Larochelle, Catherine (2021): The impact of iron-biofortified bean adoption on bean productivity, consumption, purchases and sales. In *World Development* 139, p. 105260. DOI: 10.1016/j.worlddev.2020.105260.

Reviewers' Comments:

Reviewer #2:

Remarks to the Author:

Thank you very much for the thorough response and edits based on the initial review. I agree the manuscript has improved a lot. There are a few remaining questions and recommendations.

Results:

- The key variable of production and production value as well as quantity sold and sales value are still not sufficiently defined and elaborated. It would seem that production value and sales value is just a conversion of production / sales volume into monetary values using the market price. If this is the case, there is no value added in having these as separate outcome variables. If they account for differences in the prices the individual household receives for their produce, there might be a story to be told. Line 153+154 suggest that it is not simply a linear transformation – while line 362 would suggest that it is. It might be an option to use the sales price instead as that is the underlying difference and would also lend some insights towards the conclusions drawn around transaction costs.

- The mediation analysis and its value add: It would seem that the overall result is that adoption itself is not a sufficient condition for commercialization but that the yield increasing effect of the newly adopted variety is the factor that drives the process. This is a rather obvious point and could be easily argue without the complication of the mediation analysis which for me still leaves more questions. Relatedly, the comments on non-separable agricultural household model in the response as well as in the manuscript now beg the question why home consumption has not featured in the analysis. This might be justifiable, but it would lend much more insights than the current mediation analysis to elaborate how home consumption and sales decisions are connected which I assume could be possible using the mediation analysis.

- Line 80: “quantiles of the conditional distribution of commercialization” – I assume the quantiles refer to land size or groundnut production area? If so, please correct the phrasing here and in other places where this is used.

- Line 156+157: It would be the easiest to just compare market prices for the relevant years and provide insights if the suggested increase in those is in fact behind the observed effects. The effect would however also affect non-adopters.

- Line 195: there is no analysis on the structural transformation here. I suggest to delete “and structural” from the sentence.

- Line 240-242: The different household characteristics certainly add to the understanding of the observed differences. Yet the difference in yield can not be explained by this so there appears to be other important factors at play.

- Line 297-307: This section is not supported by the analysis. The explanations in the rebuttal are fair, however it would have to be made clear that these recommendations emerge from discussing the results against the background of the wider literature. Without any references in the text it suggests that these are direct results from the analysis that has been performed which is not the case.

- Line 368: It still mentions that the area under improved groundnuts could be a proxy for specialization. I would recommend to delete “which could also be seen as a proxy for specialisation” as it is not correct.

Some sentences could still benefit from further editing. Some suggested rephrasing below to outline how things could improve. These are just suggestions of course.

- Line 14: “However, the biggest gains are more predominant among farmers who produce at smaller scales, suggesting some form of inclusivity.” One example of overcomplicating sentences. Why not simplify for example as: “However, the biggest gains are found among the smaller producers which suggest that the adoption is inclusive. ”

- Line 30 “Improving smallholder commercialization...” what is it about commercialization that is to be improved. I suggest dropping the word improving or being more precise on what aspects are to be improved.

- Line 44 “Vagaries of weather” why not use common phrases such as unpredictability of weather / increasingly erratic weather / ...

- Line 48 “ who argues constructively on ... ” constructively is the wrong term here. Barret outlines the relationship between technology adoption and markets.

- Line 54: “...to transit from semi-subsistence poverty trap.” Suggest rephrasing to ... enabling

them to further integrate into the market and potentially escape the poverty trap of semi-subsistent agriculture.

- Line 62: the reduction of inorganic fertilizer use is not a way to improve soil fertility as the sentence currently states. It "... may reduce the use of inorganic fertilizer as the legume crop itself already improves soil fertility"
- Line 66: "cash-constraining for" should be "cash-constrained"
- Line 87: "somewhat upstream segment" what is meant here?
- Line 97: "...improve learning on adopting..." change to "...improve the understanding of the implications of adopting..."
- Line 121: "...are better than non-adopters, both in terms of averages and distribution..." What is better about the distribution? It looks like the distribution has no dominance
- Line 123: replace "seems to be" with "is"
- Line 124: the reference to seed system is inappropriate as there might be many other reasons for the increase based on seed distributions by NGOs, farmer to farmer exchange, government subsidies, ...
- Line 150: replace "with" by "and"
- Line 166: Replace "highlight" with "support"
- Line 185: "first-hand outcomes" I guess "first-order outcomes" is meant here
- Line 194: "groundnut varieties" should be "groundnut variety adoption"
- Line 195: Drop "dummy"
- Line 201: The full name for ICRISAT is missing "for the Semi-Arid Tropics"
- Line 234: rephrase to "We observe the strongest yield effects in Ghana and Nigeria."
- Line 256: drop the insert "area under adoption is smaller" adopt at smaller scale is sufficient.
- Line 267: replace "like" with "similar to"
- Line 282: delete "insights employ"

Referencing:

- Line 57: The way the sentence is written still implies that both Kassie et al. and Verkaart et al.'s work is about groundnuts. I would suggest moving the reference to line 64 after environmentally friendly.
- Line 498: Michler et al. is not about groundnut. Replace "improved climate-resilient groundnut varieties" with "improved chickpea varieties"

Response to Reviewer's comments

The Adoption of Climate-Resilient Groundnut Varieties Increases Agricultural Production and Smallholder Commercialization in West Africa

Reviewer #2 (Remarks to the Author):

Thank you very much for the thorough response and edits based on the initial review. I agree the manuscript has improved a lot. There are a few remaining questions and recommendations.

Thank you for your further constructive comments and suggestions. Your comments have again helped us to improve the overall quality of the paper. Following your suggestions, we have made more revisions which we believe have improved the readability of the paper. Below, we reproduce your comments in plain text, and beneath it, we provide our replies in blue text. We hope that our revisions have fully addressed your comments. We have also submitted a file with tracked changes where we show all the corrections and edits that have been implemented in this round of revision.

Results:

- The key variable of production and production value as well as quantity sold and sales value are still not sufficiently defined and elaborated. It would seem that production value and sales value is just a conversion of production / sales volume into monetary values using the market price. If this is the case, there is no value added in having these as separate outcome variables. If they account for differences in the prices the individual household receives for their produce, there might be a story to be told. Line 153+154 suggest that it is not simply a linear transformation – while line 362 would suggest that it is. It might be an option to use the sales price instead as that is the underlying difference and would also lend some insights towards the conclusions drawn around transaction costs.

Thank you for this clarification. Yes, we indeed used the individual sales price to account for the different prices farmers were receiving. We have made this clearer in the manuscript and we agree that it is various aspects of prices that are associated with some of the discussion and conclusion about transaction costs.

- The mediation analysis and its value add: It would seem that the overall result is that adoption itself is not a sufficient condition for commercialization but that the yield increasing effect of the newly adopted variety is the factor that drives the process. This is a rather obvious point and could be easily argue without the complication of the mediation analysis which for me still leaves more questions. Relatedly, the comments on non-separable agricultural household model in the response as well as in the manuscript now beg the question why home consumption has not featured in the analysis. This might be justifiable, but it would lend much more insights than the current mediation analysis to elaborate how home consumption and sales decisions are connected which I assume could be possible using the mediation analysis.

Thank you for this comment. We have now followed your recommendation in dropping the mediation analysis as it seems complicated and not very intuitive and useful for this study. We have also added home consumption to the analysis now. We have performed simple panel fixed effect regressions where we regressed sales on both production and consumption. We observe that production and sales are positively associated while consumption and sales are negatively associated. This is in accordance with theoretical expectations especially given that groundnut is consumed by households. Given that the production and consumption decisions of households are non-separable, insights from this new regression makes intuitive sense and is supportive of previous insights that households only participate in markets to the extent that their households' demands are met. Furthermore, we have added new results where we explore the link between adoption and consumption as well. We are very grateful that you could push us to add consumption in the manuscript. The manuscript feels complete and round with this additional analysis.

- Line 80: “quantiles of the conditional distribution of commercialization” – I assume the quantiles refer to land size or groundnut production area? If so, please correct the phrasing here and in other places where this is used.

The quantiles here refer to various cut-off points in commercialization after dividing it in equal groups.

- Line 156+157: It would be the easiest to just compare market prices for the relevant years and provide insights if the suggested increase in those is in fact behind the observed effects. The effect would however also affect non-adopters.

This is right. It is for this reason that we prefer to use the prices farmers receive which capture some aspects of transaction costs which is important for commercialization.

- Line 195: there is no analysis on the structural transformation here. I suggest to delete “and structural” from the sentence.

We have deleted “structural” from the sentence.

- Line 240-242: The different household characteristics certainly add to the understanding of the observed differences. Yet the difference in yield can not be explained by this so there appears to be other important factors at play.

This is right. We have reflected this in the manuscript

- Line 297-307: This section is not supported by the analysis. The explanations in the rebuttal are fair, however it would have to be made clear that these recommendations emerge from discussing the results against the background of the wider literature. Without any references in

the text it suggests that these are direct results from the analysis that has been performed which is not the case.

We agree with this and have now mentioned that some of these insights do not result from the analysis. We have also added some citations here to show that these insights do not emerge from the analysis but are rather based on some parts of the broader literature on smallholder commercialization.

- Line 368: It still mentions that the area under improved groundnuts could be a proxy for specialization. I would recommend to delete “which could also be seen as a proxy for specialisation” as it is not correct.

We have taken out that statement.

Some sentences could still benefit from further editing. Some suggested rephrasing below to outline how things could improve. These are just suggestions of course.

Thank you for these recommendations which we have followed.

- Line 14: “However, the biggest gains are more predominant among farmers who produce at smaller scales, suggesting some form of inclusivity.” One example of overcomplicating sentences. Why not simplify for example as: “However, the biggest gains are found among the smaller producers which suggest that the adoption is inclusive.”

We have followed your recommendation.

- Line 30 “Improving smallholder commercialization... “ what is it about commercialization that is to be improved. I suggest dropping the word improving or being more precise on what aspects are to be improved.

We have followed your recommendation.

- Line 44 “Vagaries of weather” why not use common phrases such as unpredictability of weather / increasingly erratic weather / ...

We have followed your recommendation.

- Line 48 “ who argues constructively on ... “ constructively is the wrong term here. Barret outlines the relationship between technology adoption and markets.

We have followed your recommendation.

- Line 54: “...to transit from semi-subsistence poverty trap.” Suggest rephrasing to ... enabling them to further integrate into the market and potentially escape the poverty trap of semi-subsistent agriculture.

We have followed your recommendation.

- Line 62: the reduction of inorganic fertilizer use is not a way to improve soil fertility as the sentence currently states. It "... may reduce the use of inorganic fertilizer as the legume crop itself already improves soil fertility"

We have followed your recommendation.

- Line 66: "cash-constraining for" should be "cash-constrained"

We have followed your recommendation.

- Line 87: "somewhat upstream segment" what is meant here?

By upstream segment, we refer to production factors. We have now qualified by adding "production" to the sentence.

- Line 97: "...improve learning on adopting..." change to "...improve the understanding of the implications of adopting..."

We have followed your recommendation.

- Line 121: "...are better than non-adopters, both in terms of averages and distribution..." What is better about the distribution? It looks like the distribution has no dominance

We have taken out "distribution" from the sentence.

- Line 123: replace "seems to be" with "is"

We have followed your recommendation.

- Line 124: the reference to seed system is inappropriate as there might be many other reasons for the increase based on seed distributions by NGOs, farmer to farmer exchange, government subsidies, ...

We have followed your recommendation and added some of these other factors.

- Line 150: replace "with" by "and"

We have followed your recommendation.

- Line 166: Replace “highlight” with “support”

We have followed your recommendation.

- Line 185: “first-hand outcomes” I guess “first-order outcomes” is meant here
We have followed your recommendation.

- Line 194: “groundnut varieties” should be “groundnut variety adoption”

We have followed your recommendation.

- Line 195: Drop “dummy”

We have followed your recommendation.

- Line 201: The full name for ICRISAT is missing “for the Semi-Arid Tropics”

We have followed your recommendation.

- Line 234: rephrase to “We observe the strongest yield effects in Ghana and Nigeria.”

We have followed your recommendation.

- Line 256: drop the insert “area under adoption is smaller” adopt at smaller scale is sufficient.

We have followed your recommendation.

- Line 267: replace “like” with “similar to”

We have followed your recommendation.

- Line 282: delete “insights employ”

We have followed your recommendation.

Referencing:

- Line 57: The way the sentence is written still implies that both Kassie et al. and Verkaart et al.’s work is about groundnuts. I would suggest moving the reference to line 64 after environmentally friendly.

We have followed your recommendation.

- Line 498: Michler et al. is not about groundnut. Replace “improved climate-resilient groundnut varieties” with “improved chickpea varieties”

We have followed your recommendation.

Reviewers' Comments:

Reviewer #2:

Remarks to the Author:

Thank you for the careful consideration of the concerns raised. I am now satisfied with the manuscript.